# Does mathematics training lead to better logical thinking and reasoning? A cross-sectional assessment from students to professors

Clio Cresswell[1], Craig P. Speelman[2]*

**1** School of Mathematics and Statistics, The University of Sydney, Sydney, Australia, **2** School of Arts and Humanities, Edith Cowan University, Joondalup, Australia

* c.speelman@ecu.edu.au

## Abstract

Mathematics is often promoted as endowing those who study it with transferable skills such as an ability to think logically and critically or to have improved investigative skills, resourcefulness and creativity in problem solving. However, there is scant evidence to back up such claims. This project tested participants with increasing levels of mathematics training on 11 well-studied rational and logical reasoning tasks aggregated from various psychological studies. These tasks, that included the Cognitive Reflection Test and the Wason Selection Task, are of particular interest as they have typically and reliably eluded participants in all studies, and results have been uncorrelated with general intelligence, education levels and other demographic information. The results in this study revealed that in general the greater the mathematics training of the participant, the more tasks were completed correctly, and that performance on some tasks was also associated with performance on others not traditionally associated. A ceiling effect also emerged. The work is deconstructed from the viewpoint of adding to the platform from which to approach the greater, and more scientifically elusive, question: are any skills associated with mathematics training innate or do they arise from skills transfer?

**Data Availability Statement:** All relevant data are within the paper and its Supporting Information files.

## Introduction

Mathematics is often promoted as endowing those who study it with a number of broad thinking skills such as: an ability to think logically, analytically, critically and abstractly; having capacity to weigh evidence with impartiality. This is a view of mathematics as providing transferable skills which can be found across educational institutions, governments and corporations worldwide. A view material to the place of mathematics in curricula.

Consider the UK government's commissioned inquiry into mathematics education "Making Mathematics Count" ascertaining the justification that "mathematical training disciplines the mind, develops logical and critical reasoning, and develops analytical and problem-solving skills to a high degree" [1 p11]. The Australian Mathematical Sciences Institute very broadly

**Funding:** The authors received no specific funding for this work.

**Competing interests:** The authors have declared that no competing interests exist.

states in its policy document "Vision for a Maths Nation" that "Not only is mathematics *the* enabling discipline, it has a vital productive role planning and protecting our well-being" (emphasis in original) [2]. In Canada, British Columbia's New 2016 curriculum K-9 expressly mentions as part of its "Goals and Rationale": "The Mathematics program of study is designed to develop deep mathematical understanding and fluency, logical reasoning, analytical thought, and creative thinking." [3]. Universities, too, often make such specific claims with respect to their teaching programs. "Mathematics and statistics will help you to think logically and clearly, and apply a range of problem-solving strategies" is claimed by The School of Mathematical Sciences at Monash University, Australia [4]. The School of Mathematics and Statistics at The University of Sydney, Australia, directly attributes as part of particular course objectives and outcomes skills that include "enhance your problem-solving skills" as part of studies in first year [5], "develop logical thinking" as part of studies in second year, which was a statement drafted by the lead author in fact [6], and "be fluent in analysing and constructing logical arguments" as part of studies in third year [7]. The University of Cambridge's Faculty of Mathematics, UK, provides a dedicated document "Transferable Skills in the Mathematical Tripos" as part of its undergraduate mathematics course information, which again lists "analytic ability; creativity; initiative; logical and methodical reasoning; persistence" [8].

In contrast, psychological research, which has been empirically investigating the concept of transferability of skills since the early 1900s, points quite oppositely to reasoning skills as being highly domain specific [9]. Therefore, support for claims that studying mathematics engenders more than specific mathematics knowledge is highly pertinent. And yet it is largely absent. The 2014 Centre for Curriculum Redesign (CCR) four part paper "Mathematics for the 21st Century: What Should Students Learn?" concludes in its fourth paper titled "Does mathematics education enhance higher-order thinking skills?" with a call to action ". . . there is not sufficient evidence to conclude that mathematics enhances higher order cognitive functions. The CCR calls for a much stronger cognitive psychology and neuroscience research base to be developed on the effects of studying mathematics" [10].

Inglis and Simpson [11], bringing up this very issue, examined the ability of first-year undergraduate students from a high-ranking UK university mathematics department, on the "Four Cards Problem" thinking task, also known as the Wason Selection Task. It is stated as follows.

Each of the following cards have a letter on one side and a number on the other.

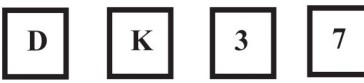

Here is a rule: "if a card has a D on one side, then it has a 3 on the other". Your task is to select all those cards, but only those cards, which you would have to turn over in order to find out whether the rule is true or false. Which cards would you select?

This task involves understanding conditional inference, namely understanding the rule "If P then Q" and with this, deducing the answer as "P and not Q" or "D and 7". Such logical deduction indeed presents as a good candidate to test for a potential ability of the mathematically trained. This task has also been substantially investigated in the domain of the psychology of reasoning [12 p8] revealing across a wide range of publications that only around 10% of the general population reach the correct result. The predominant mistake being to pick "D and 3"; where in the original study by Wason [13] it is suggested that this was picked by 65% of people. This poor success rate along with a standard mistake has fuelled interest in the task as well as

attempts to understand why it occurs. A prevailing theory being the so named matching bias effect; the effect of disproportionately concentrating on items specifically mentioned in the situation, as opposed to reasoning according to logical rules.

Inglis and Simpson's results isolated mathematically trained individuals with respect to this task. The participants were under time constraint and 13% of the first-year undergraduate mathematics students sampled reached the correct response, compared to 4% of the non-mathematics (arts) students that was included. Of note also was the 24% of mathematics students as opposed to 45% of the non-mathematics students who chose the standard mistake. The study indeed unveiled that mathematically trained individuals were significantly less affected by the matching bias effect with this problem than the individuals without mathematics training. However, the achievement of the mathematically trained group was still far from masterful and the preponderance for a non-standard mistake compared with non-mathematically trained people is suggestive. Mathematical training appears to engender a different thinking style, but it remains unclear what the difference is.

Inglis, Simpson and colleagues proceeded to follow up their results with a number of studies concentrated on conditional inference in general [14, 15]. A justification for this single investigatory pathway being that if transfer of knowledge is present, something subtle to test for in the first place, a key consideration should be the generalisation of learning rather than the application of skills learned in one context to another (where experimenter bias in the choice of contexts is more likely to be an issue). For this they typically used sixteen "if P then Q" comprehension tasks, where their samples across a number of studies have included 16-year-old pre-university mathematics students (from England and Cyprus), mathematics honours students in their first year of undergraduate university study, third year university mathematics students, and associated control groups. The studies have encompassed controls for general intelligence and thinking disposition prior to training, as well as follows ups of up to two years to address the issue of causation. The conclusive thinking pattern that has emerged is a tendency of the mathematical groups towards a greater likelihood of rejecting the invalid denial of the antecedent and affirmation of the consequent inferences. But with this, and this was validated by a second separate study, the English mathematics group actually became less likely to endorse the valid modus tollens inference. So again, mathematical training appears to engender a different thinking style, but there are subtleties and it remains unclear what the exact difference is.

This project was designed to broaden the search on the notion that mathematics training leads to increased reasoning skills. We focused on a range of reasoning problems considered in psychological research to be particularly insightful into decision making, critical thinking and logical deduction, with their distinction in that the general population generally struggles with answering them correctly. An Australian sample adds diversity to the current enquiries that have been European focussed. Furthermore, in an effort to identify the impact of mathematics training through a possible gradation effect, different levels of mathematically trained individuals were tested for performance.

## The study

Well-studied thinking tasks from a variety of psychological studies were chosen. Their descriptions, associated success rates and other pertinent details follows. They were all chosen as the correct answer is typically eluded for a standard mistake.

The three-item *Cognitive Reflection Test* (CRT) was used as introduced by Frederick [16]. This test was devised in line with the theory that there are two general types of cognitive activity: one that operates quickly and without reflection, and another that requires not only

conscious thought and effort, but also an ability to reflect on one's own cognition by including a step of suppression of the first to reach it. The three items in the test involve an incorrect "gut" response and further cognitive skill is deemed required to reach the correct answer (although see [17] for evidence that correct responses can result from "intuition", which could be related to intelligence [18]).

### Lily pads

In a lake, there is a patch of lily pads. Every day, the patch doubles in size. If it takes 48 days for the patch to cover the entire lake, how long would it take for the patch to cover half of the lake?

### Widgets

If it takes 5 machines 5 minutes to make 5 widgets, how long would it take 100 machines to make 100 widgets?

### Bat and ball

A bat and a ball cost $1.10 in total. The bat costs a dollar more than the ball. How much does the ball cost?

The solutions are: 47 days for the Lily Pads problem, 5 minutes for the Widgets problem and 5 cents for the Bat and Ball problem. The considered intuitive, but wrong, answers are 24 days, 100 minutes and 10 cents, respectively. These wrong answers are attributed to participants becoming over focused on the numbers so as to ignore the exponential growth pattern in the Lily Pads problem, merely complete a pattern in numbers in the Widgets problem, and neglect the relationship "more than" in the Bat and Ball problem [19]. The original study by Frederick [16] provides a composite measure of the performance on these three items, with only 17% of those studied (n = 3428) reaching the perfect score. The CRT has since been studied extensively [19–21]. Research using the CRT tends not to report performance on the individual items of the test, but rather a composite measure of performance. Attridge and Inglis [22] used the CRT as a test for thinking disposition of mathematics students as one way to attempt to disentangle the issue of filtering according to prior thinking styles rather than transference of knowledge in successful problem solving. They repeat tested 16-year old pre-university mathematics students and English literature students without mathematics subjects at a one-year interval and found no difference between groups.

Three problems were included that test the ability to reason about probability. All three problems were originally discussed by Kahneman and Tversky [23], with the typically poor performance on these problems explained by participants relying not on probability knowledge, but a short-cut method of thinking known as the representativeness heuristic. In the late 1980s, Richard Nisbett and colleagues showed that graduate level training in statistics, while not revealing any improvement in logical reasoning, did correlate with higher-quality statistical answers [24]. Their studies lead in particular to the conclusion that comprehension of, what is known as the law of large numbers, did show improvement with training. The first of our next three problems targeted this law directly.

### Hospitals

A certain town is served by two hospitals. In the larger hospital, about 45 babies are born each day, and in the smaller hospital, about 15 babies are born each day. As you know, about 50 percent of all babies are boys. However, the exact percentage varies from day to day. Sometimes it

may be higher than 50 percent, sometimes lower. For a period of one year, each hospital recorded the number of days on which more than 60 percent of the babies born were boys. Which hospital do you think recorded more such days? (Circle one letter.)

(a). the larger hospital

(b). the smaller hospital

(c). about the same (that is, within 5 percent of each other)

Kahneman and Tversky [23] reported that, of 50 participants, 12 chose (a), 10 chose (b), and 28 chose (c). The correct answer is (b), for the reason that small samples are more likely to exhibit extreme events than large samples from the same population. The larger the sample, the more likely it will exhibit characteristics of the parent population, such as the proportion of boys to girls. However, people tend to discount or be unaware of this feature of sampling statistics, which Kahneman and Tversky refer to as the law of large numbers. Instead, according to Kahneman and Tversky, people tend to adhere to a fallacious law of small numbers, where even small samples are expected to exhibit properties of the parent population, as illustrated by the proportion of participants choosing the answer (c) in their 1972 study. Such thinking reflects use of the representativeness heuristic, whereby someone will judge the likelihood of an uncertain event based on how similar it is to characteristics of the parent population of events.

### Birth order

All families of six children in a city were surveyed. In 72 families the exact order of births of boys and girls was GBGBBG.

(a). What is your estimate of the number of families surveyed in which the exact order of births was BGBBBB?

(b). In the same survey set, which, if any, of the following two sequences would be more likely: BBBGGG or GBBGBG?

All of the events listed in the problem have an equal probability, so the correct answer to (a) is 72, and to (b) is "neither is more likely". Kahneman and Tversky [23] reported that 75 of 92 participants judged the sequence in (a) as less likely than the given sequence. A similar number (unspecified by Kahneman and Tversky, but the statistical effect was reported to be of the same order as in (a)) reported that GBBGBG was the more likely sequence. Again, Kahneman and Tversky suggested that these results reflected use of the representativeness heuristic. In the context of this problem, the heuristic would have taken the following form: some birth orders appear less patterned than others, and less patterned is to be associated with the randomness of birth order, making them more likely.

### Coin tosses

In a sequence of coin tosses (the coin is fair) which of the following outcomes would be most likely (circle one letter):

(a). H T H T H T H T

(b). H H H H T T T T

(c). T T H H T T H H

(d).  H T T H T H H T

(e).  all of the above are equally likely

The correct answer in this problem is (e). Kahneman and Tversky [23] reported that participants tend to choose less patterned looking sequences (e.g., H T T H T H H T) as more likely than more systematic looking sequences (e.g., H T H T H T H T). This reasoning again reflects the representativeness heuristic.

Three further questions from the literature were included to test problem solving skill.

## Two drivers

Two drivers set out on a 100-mile race that is marked off into two 50-mile sections. Driver A travels at exactly 50 miles per hour during the entire race. Driver B travels at exactly 45 mph during the first half of the race (up to the 50-mile marker) and travels at exactly 55 mph during the last half of the race (up to the finish line). Which of the two drivers would win the race? (Circle one letter.)

(a).  Driver A would win the race

(b).  Driver B would win the race

(c).  the two drivers would arrive at the same time (within a few seconds of one another)

This problem was developed by Pelham and Neter [25]. The correct answer is (a), which can be determined by calculations of driving times for each Driver, using time = distance/ velocity. Pelham and Neter argue, however, that (c) is intuitively appealing, on the basis that both drivers appear to have the same overall average speed. Pelham and Neter reported that 67% of their sample gave this incorrect response to the problem, and a further 13% selected (b).

## Petrol station

Imagine that you are driving along the road and you notice that your car is running low on petrol. You see two petrol stations next to each other, both advertising their petrol prices. Station A's price is 65c/litre; Station B's price is 60c/litre. Station A's sign also announces: "5c/litre discount for cash!" Station B's sign announces "5c/litre surcharge for credit cards." All other factors being equal (for example, cleanliness of the stations, number of cars waiting at each etc), to which station would you choose to go, and why?

This problem was adapted from one described by Galotti [26], and is inspired by research reported by Thaler [27]. According to Thaler's research, most people prefer Station A, even though both stations are offering the same deal: 60c/litre for cash, and 65c/litre for credit. Tversky and Kahneman [28] explain this preference by invoking the concept of framing effects. In the context of this problem, such an effect would involve viewing the outcomes as changes from some initial point. The initial point frames the problem, and provides a context for viewing the outcome. Thus, depending on the starting point, outcomes in this problem can be viewed as either a gain (in Station A, you gain a discount if you use cash) or a loss (in Station B, you are charged more (a loss) for using credit). Given that people are apparently more concerned about a loss than a gain [29], the loss associated with Station B makes it the less attractive option, and hence the preference for Station A. The correct answer, though, is that the stations are offering the same deal and so no station should be preferred.

And finally, a question described by Stanovich [30, 31] as testing our predisposition for cognitive operations that require the least computational effort.

### Jack looking at Anne

Jack is looking at Anne, but Anne is looking at George. Jack is married, but George is not. Is a married person looking at an unmarried person? (Circle one letter.)

(a). Yes

(b). No

(c). Cannot be determined

Stanovich reported that over 80% of people choose the "lazy" answer (c). The correct answer is (a).

The above questions survey, in a clear problem solving setting, an ability to engage advanced cognitive processing in order to critically evaluate and possibly override initial gut reasoning, an ability to reason about probability within the framework of the law of large numbers and the relationship between randomness and patterning, an ability to isolate salient features of a problem and, with the last question in particular, an ability to map logical relations. It might be hypothesised that according to degrees of mathematical training, in line with the knowledge base provided and the claims of associated broad and enhanced problem-solving abilities in general, that participants with greater degrees of such training would outperform others on these questions. This hypothesis was investigated in this study. In addition, given that no previous study on this issue has examined the variety of problems used in this study, we also undertook an exploratory analysis to investigate whether there exist any associations between the problems in terms of their likelihood of correct solution. Similarities between problems might indicate which problem solving domains could be susceptible to the effects of mathematics training.

## Method

### Design

A questionnaire was constructed containing the problems described in the previous sections plus the Four Cards Problem as tested by Inglis and Simpson [11] for comparison. The order of the problems was as follows: 1) Lily Pads; 2) Hospitals; 3) Widgets; 4) Four Cards; 5) Bat and Ball; 6) Birth Order; 7) Petrol Station; 8) Coin Tosses; 9) Two Drivers; 10) Jack looking at Anne. It was administered to five groups distinctive in mathematics training levels chosen from a high-ranking Australian university, where the teaching year is separated into two teaching semesters and where being a successful university applicant requires having been highly ranked against peers in terms of intellectual achievement:

1. Introductory—First year, second semester, university students with weak high school mathematical results, only enrolled in the current unit as a compulsory component for their chosen degree, a unit not enabling any future mathematical pathway, a typical student may be enrolled in a Biology or Geography major;

2. Standard—First year, second semester, university students with fair to good high school mathematical results, enrolled in the current mathematics unit as a compulsory component for their chosen degree with the possibility of including some further mathematical units in their degree pathway, a typical student may be enrolled in an IT or Computer Science major;

3. Advanced1—First year, second semester, university mathematics students with very strong interest as well as background in mathematics, all higher year mathematical units are

included as possible future pathway, a typical student may be enrolled in a Mathematics or Physics major;

4. Advanced2—Second year, second semester, university mathematics students with strong interest as well as background in mathematics, typically a direct follow on from the previously mentioned Advanced1 cohort;

5. Academic—Research academics in the mathematical sciences.

## Participants

123 first year university students volunteered during "help on demand" tutorial times containing up to 30 students. These are course allocated times that are supervised yet self-directed by students. This minimised disruption and discouraged coercion. 44 second year university students completed the questionnaire during a weekly one-hour time slot dedicated to putting the latest mathematical concepts to practice with the lecturer (whereby contrast to what occurs in tutorial times the lecturer does most of the work and all students enrolled are invited). All these university students completed the questionnaire in normal classroom conditions; they were not placed under strict examination conditions. The lead author walked around to prevent discussion and coercion and there was minimum disruption. 30 research academics responded to local advertising and answered the questionnaire in their workplace while supervised.

## Procedure

The questionnaires were voluntary, anonymous and confidential. Participants were free to withdraw from the study at any time and without any penalty. No participant took this option however. The questionnaires gathered demographic information which included age, level of education attained and current qualification pursued, name of last qualification and years since obtaining it, and an option to note current speciality for research academics. Each problem task was placed on a separate page. Participants were not placed under time constraint, but while supervised, were asked to write their start and finish times on the front page of the survey to note approximate completion times. Speed of completion was not incentivised. Participants were not allowed to use calculators. A final "Comments Page" gave the option for feedback including specifically if the participants had previously seen any of the questions. Questionnaires were administered in person and supervised to avoid collusion or consulting of external sources.

The responses were coded four ways: A) correct; B) standard error (the errors discussed above in The Study); C) other error; D) left blank.

The ethical aspects of the study were approved by the Human Research Ethics Committee of the University of Sydney, protocol number [2016/647].

## Results

The first analysis examined the total number of correct responses provided by the participants as a function of group. Scores ranged from 1 to 11 out of a total possible of 11 (Problem 6 had 2 parts) (Fig 1). An ANOVA of this data indicated a significant effect of group ($F(4, 192) = 20.426$, $p < .001$, partial $\eta^2 = .299$). Pairwise comparisons using Tukey's HSD test indicated that the Introductory group performed significantly worse than the Advanced1, Advanced2 and Academic groups. There were no significant differences between the Advanced1, Advanced2 and Academic groups.

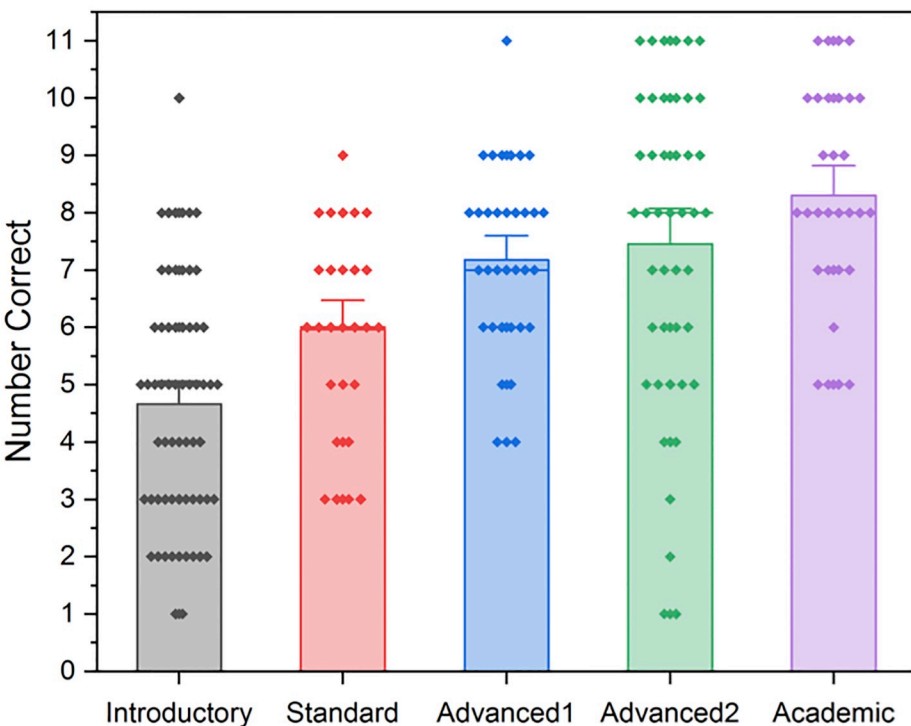

**Fig 1. Mean number of correct responses (numbcorr) for each group.** Error bars are one standard error of the mean.

Overall solution time, while recorded manually and approximately, was positively correlated with group, such that the more training someone had received, the longer were these solution times (r(180) = 0.247, p = .001). However, as can be seen in Fig 2, this relationship is not strong.

A series of chi-squared analyses, and their Bayesian equivalents, were performed on each problem, to determine whether the distribution of response types differed as a function of group. To minimise the number of cells in which expected values in some of these analyses were less than 5, the Standard Error, Other Error and Blank response categories were collapsed into one category (Incorrect Response). For three of the questions, the expected values of some cells did fall below 5, and this was due to most people getting the problem wrong (Four Cards), or most people correctly responding to the problem (Bat and Ball, Coin Tosses). In these cases, the pattern of results was so clear that a statistical analysis was barely required. Significant chi-squared results were examined further with pairwise posthoc comparisons (see Table 1).

## Four cards

The three groups with the least amount of training in mathematics were far less likely than the other groups to give the correct solution ($\chi^2$ (4) = 31.06, p < .001; $BF_{10}$ = 45,045) (Table 1). People in the two most advanced groups (Advanced2 and Academic) were more likely to solve the card problem correctly, although it was still less than half of the people in these groups who did so. Further, these people were less likely to give the standard incorrect solution, so that most who were incorrect suggested some more cognitively elaborate answer, such as turning over all cards. The proportion of people in the Advanced2 and Academic groups (39 and 37%) who solved the problem correctly far exceeded the typical proportion observed with this problem (10%). Of note, also, is the relatively high proportion of those in the higher training groups

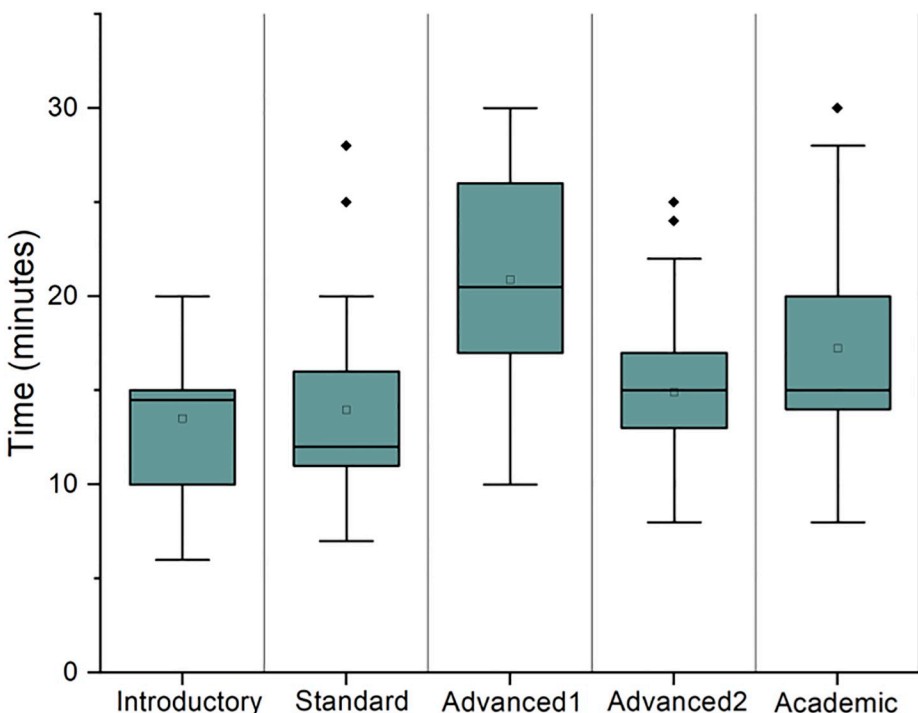

**Fig 2. Box and whisker diagrams depicting mean approximate response time (minutes) for each group.**

who, when they made an error, did not make the standard error, a similar result to the one reported by Inglis and Simpson [11].

### The cognitive reflection test

In the Lily Pads problem, although most people in the Standard, Advanced1, Advanced2 and Academic groups were likely to select the correct solution, it was also the case that the less training someone had received in mathematics, the more likely they were to select an incorrect solution ($\chi^2$ (4) = 27.28, p < .001; $BF_{10}$ = 15,554), with the standard incorrect answer being the next most prevalent response for the two lower ability mathematics groups (Table 1).

Performance on the Widgets problem was similar to performance on the Lily Pads problem in that most people in the Standard, Advanced1, Advanced2 and Academic groups were likely to select the correct solution, but that the less training someone had received in mathematics, the more likely they were to select an incorrect solution ($\chi^2$ (4) = 23.76, p< .001; $BF_{10}$ = 516) (Table 1). As with the Lily Pads and Widget problems, people in the Standard, Advanced1, Advanced2 and Academic groups were highly likely to solve the Bat and Ball problem ($\chi^2$ (4) = 35.37, p < .001; $BF_{10}$ = 208,667). Errors were more likely from the least mathematically trained people (Introductory, Standard) than the other groups (Table 1).

To compare performance on the CRT with previously published results, performance on the three problems (Lily Pads, Widgets, Bat and Ball) were combined. The number of people in each condition that solved 0, 1, 2, or 3 problems correctly is presented in Table 2. The Introductory group were evenly distributed amongst the four categories, with 26% solving all three problems correctly. Around 70% of the rest of the groups solved all 3 problems correctly, which is vastly superior to the 17% reported by Frederick [16].

**Table 1. Proportion of participants selecting particular response types for all problems as a function of group.**

|  | Introductory[a] | Standard[b] | Advanced1[c] | Advanced2[d] | Academic[e] |
|---|---|---|---|---|---|
| n | 62 | 27 | 34 | 44 | 30 |
| Mean Age (yrs) | 19.3 | 18.9 | 18.6 | 20.0 | 40.2 |
| Response Type |  |  |  |  |  |
| *Cards* | d,e | d,e |  | a,b | a,b |
| Correct | 5% | 4% | 12% | 39% | 37% |
| Standard Error | 47% | 41% | 21% | 18% | 7% |
| Other Error | 45% | 52% | 65% | 43% | 57% |
| Blank | 3% | 4% | 3% | 0% | 0% |
| *Lily Pads* | c,d |  | a | a |  |
| Correct | 55% | 78% | 85% | 82% | 100% |
| Standard Error | 29% | 11% | 3% | 9% | 0% |
| Other Error | 15% | 7% | 12% | 9% | 0% |
| Blank | 2% | 4% | 0% | 0% | 0% |
| *Widgets* | c,d,e |  | a | a | a |
| Correct | 48% | 74% | 82% | 82% | 87% |
| Standard Error | 31% | 11% | 15% | 11% | 7% |
| Other Error | 18% | 15% | 3% | 7% | 7% |
| Blank | 3% | 0% | 0% | 0% | 0% |
| *Bat & Ball* |  |  |  |  |  |
| Correct | 60% | 89% | 100% | 93% | 90% |
| Standard Error | 31% | 11% | 0% | 5% | 7% |
| Other Error | 8% | 0% | 0% | 2% | 0% |
| Blank | 2% | 0% | 0% | 0% | 3% |
| *Hospitals* |  |  |  |  |  |
| Correct | 32% | 48% | 44% | 52% | 47% |
| Standard Error | 48% | 44% | 53% | 43% | 50% |
| Other Error | 18% | 7% | 3% | 5% | 3% |
| Blank | 2% | 0% | 0% | 0% | 0% |
| *Birth Order (a)* | c,d,e |  | a | a | a |
| Correct | 35% | 52% | 71% | 66% | 83% |
| Standard Error | 35% | 22% | 18% | 9% | 13% |
| Other Error | 16% | 15% | 9% | 16% | 3% |
| Blank | 13% | 11% | 3% | 9% | 0% |
| *Birth Order (b)* | d |  |  | a |  |
| Correct | 53% | 63% | 88% | 82% | 93% |
| Standard Error | 34% | 19% | 9% | 14% | 3% |
| Other Error | 3% | 7% | 0% | 2% | 3% |
| Blank | 10% | 11% | 3% | 2% | 0% |
| *Coin Tosses* |  |  |  |  |  |
| Correct | 92% | 96% | 91% | 84% | 93% |
| Standard Error | 3% | 4% | 3% | 5% | 0% |
| Other Error | 3% | 0% | 6% | 11% | 7% |
| Blank | 2% | 0% | 0% | 0% | 0% |
| *Two Drivers* | c,d,e | c,d,e | a,b | a,b | a,b |
| Correct | 16% | 19% | 56% | 64% | 73% |
| Standard Error | 71% | 67% | 29% | 25% | 23% |
| Other Error | 11% | 15% | 15% | 11% | 3% |

*(Continued)*

**Table 1.** (Continued)

| | Introductory[a] | Standard[b] | Advanced1[c] | Advanced2[d] | Academic[e] |
|---|---|---|---|---|---|
| n | 62 | 27 | 34 | 44 | 30 |
| Mean Age (yrs) | 19.3 | 18.9 | 18.6 | 20.0 | 40.2 |
| Blank | 2% | 0% | 0% | 0% | 0% |
| *Petrol Station* | | | | | |
| Correct | 50% | 56% | 59% | 55% | 73% |
| Standard Error | 27% | 22% | 18% | 25% | 3% |
| Other Error | 21% | 22% | 24% | 20% | 20% |
| Blank | 2% | 0% | 0% | 0% | 3% |
| *Jack looking at Anne* | d,e | | | a | a |
| Correct | 19% | 19% | 29% | 48% | 53% |
| Standard Error | 71% | 81% | 68% | 52% | 47% |
| Other Error | 6% | 0% | 3% | 0% | 0% |
| Blank | 3% | 0% | 0% | 0% | 0% |

Superscripts label the groups (e.g., Introductory = a). Within the table, these letters refer to which other group a particular group was significantly different to according to a series of pairwise post hoc chi squared analyses (Bonferroni corrected $\alpha$ = .005) (e.g., 'd' in the Introductory column indicates the Introductory and the Advanced2 (d) group were significantly different for a particular problem).

## Hospitals

Responses to the Hospitals problem were almost universally split between correct and standard errors in the Standard, Advanced1, Advanced2 and Academic groups. Although this pattern of responses was also evident in the Introductory group, this group also exhibited more non-standard errors and non-responses than the other groups. However, the differences between the groups were not significant ($\chi^2$ (4) = 4.93, p = .295; $BF_{10}$ = .068) (Table 1). Nonetheless, the performance of all groups exceeds the 20% correct response rate reported by Kahneman and Tversky [23].

## Birth order

The two versions of the Birth Order problem showed similar results, with correct responses being more likely in the groups with more training (i.e., Advanced1, Advanced2 and Academic), and responses being shared amongst the various categories in the Introductory and Standard groups ($\chi_a^2$ (4) = 24.54, p < .001; $BF_{10}$ = 1,303; $\chi_b^2$ (4) = 25.77, p < .001; $BF_{10}$ = 2,970) (Table 1). Nonetheless, performance on both versions of the problem in this study was significantly better than the 82% error rate reported by Kahneman and Tversky [23].

**Table 2. Performance on the Cognitive Reflection Test as a function of group.**

| | Introductory | Standard | Advanced1 | Advanced2 | Academic |
|---|---|---|---|---|---|
| N | 62 | 27 | 34 | 44 | 30 |
| Number correct | | | | | |
| 0 | 21% | 0% | 0% | 0% | 0% |
| 1 | 21% | 30% | 3% | 14% | 0% |
| 2 | 32% | 0% | 26% | 16% | 23% |
| 3 | 26% | 70% | 71% | 70% | 77% |

## Coin tosses

The Coin Tosses problem was performed well by all groups, with very few people in any condition committing errors. There were no obvious differences between the groups ($\chi^2$ (4) = 3.70, p = .448; $BF_{10}$ = .160) (Table 1). Kahneman and Tversky [23] reported that people tend to make errors on this type of problem by choosing less patterned looking sequences, but they did not report relative proportions of people making errors versus giving correct responses. Clearly the sample in this study did not perform like those in Kahneman and Tversky's study.

## Two drivers

Responses on the Two Drivers problem were clearly distinguished by a high chance of error in the Introductory and Standard groups (over 80%), and a fairly good chance of being correct in the Advanced1, Advanced2 and Academic groups ($\chi^2$ (4) = 46.16, p < .001; $BF_{10}$ = 1.32 x $10^8$) (Table 1). Academics were the standout performers on this problem, although over a quarter of this group produced an incorrect response. Thus, the first two groups performed similarly to the participants in the Pelham and Neter [25] study, 80% of whom gave an incorrect response.

## Petrol station

Responses on the Petrol Station problem were marked by good performance by the Academic group (73% providing a correct response), and just over half of each of the other groups correctly solving the problem. This difference was not significant ($\chi^2$ (4) = 4.68, p = .322: $BF_{10}$ = .059) (Table 1). Errors were fairly evenly balanced between standard and other, except for the Academic group, who were more likely to provide a creative answer if they made an error. Thaler [27] reported that most people get this problem wrong. In this study, however, on average, most people got this problem correct, although this average was boosted by the Academic group.

## Jack looking at Anne

Responses on the Jack looking at Anne problem generally were standard errors, except for the Advanced2 and Academic groups, which were evenly split between standard errors and correct responses ($\chi^2$ (4) = 18.03, p = .001; $BF_{10}$ = 46) (Table 1). Thus, apart from these two groups, the error rate in this study was similar to that reported by Stanovich [30], where 80% of participants were incorrect.

A series of logistic regression analyses were performed in order to examine whether the likelihood of solving a particular problem correctly could be predicted on the basis of whether other problems were solved correctly. Each analysis involved selecting performance (correct or error) on one problem as the outcome variable, and performance on the other problems as predictor variables. Training (amount of training) was also included as a predictor variable in each analysis. A further logistic regression was performed with training as the outcome variable, and performance on all of the problems as predictor variables. The results of these analyses are summarised in Table 3. There were three multi-variable relationships observed in these analyses, which can be interpreted as the likelihood of solving one problem in each group being associated with solving the others in the set. These sets were: (1) Lily Pads, Widgets and Petrol Station; (2) Hospitals, Four Cards and Two Drivers; (3) Birth Order and Coin Tosses. Training also featured in each of these sets, moderating the relationships as per the results presented above for each problem.

**Table 3. Logistic regression analyses, using performance on all problems (predictor variables), except one (outcome variable), to predict performance on the outcome variable.**

| | Predictor Variables | | | | | | | | | | | | | |
|---|---|---|---|---|---|---|---|---|---|---|---|---|---|---|
| | P1 | P2 | P3 | P4 | P5 | P6a | P6b | P7 | P8 | P9 | P10 | Training | P | % |
| Outcome Variables | | | | | | | | | | | | | | |
| P1 | - | | ✓ | | | | | ✓ | | | | ✓ | < .001 | 83.8 |
| P2 | | - | | ✓ | | | | | | | | | .030 | 64.5 |
| P3 | ✓ | | - | | | | | ✓ | | | | | < .001 | 80.2 |
| P4 | | ✓ | | - | | | | | | ✓ | | ✓ | < .001 | 85.3 |
| P5 | | | | | - | | | | | | | ✓ | < .001 | 83.8 |
| P6a | | | | | | - | ✓ | | | | | | < .001 | 78.2 |
| P6b | | | | | | ✓ | - | | ✓ | | | ✓ | < .001 | 79.2 |
| P7 | ✓ | | | | | | | - | | | | | < .001 | 74.1 |
| P8 | | | | | | | ✓ | | - | | | ✓ | .003 | 91.4 |
| P9 | | | | ✓ | | | | | | - | | ✓ | < .001 | 75.6 |
| P10 | | | | | | | | | | | - | | .001 | 74.1 |
| training* | ✓ | | | ✓ | ✓ | | ✓ | | ✓ | ✓ | | - | < .001 | - |

P = Problem (1 = Four Cards; 2 = Lily Pads; 3 = Widgets; 4 = Bat & Ball; 5 = Hospitals; 6a = Birth Order (a); 6b = Birth Order (b); 7 = Coin Tosses; 8 = Two Drivers;
9 = Petrol Station; 10 = Jack looking at Anne).

training = Amount of training condition.

p = significance level of logistic regression model.

% = percentage of cases correctly classified by the logistic regression model.

✓ = significant predictor, α < .05.

* = logistic regression for the training outcome variable is multinomial, whereas all other logistic regressions are binomial.

The final "Comments Page" revealed the participants as overwhelmingly enjoying the questions. Any analysis of previous exposure to the tasks proved impossible as there was little to no alignment on participant's degree of recall, if any, and even perceptions of what exposure entailed. For example, some participants confused being exposed to the particular tasks with being habitually exposed to puzzles, or even mathematics problems, more broadly.

## Discussion

In general, the amount of mathematics training a group had received predicted their performance on the overall set of problems. The greater the training, the more problems were answered correctly, and the slower the recorded response times. There was not an obvious difference between the Advanced1, Advanced2 and Academic groups on either of these measures, however there were clear differences between this group and the Introductory and Standard groups, with the former exhibiting clearly superior accuracy. While time records were taken approximately, so as to avoid adding time pressure as a variable, that the Advanced1, Advanced2 and Academic groups recorded more time in their consideration of the problems, may suggest a "pause and consider" approach to such problems is a characteristic of the advanced groups. This is in line with what was suggested by an eye-movement tracking study of mathematically trained students attempting the Four Cards Problem; where participants that had not chosen the standard error had spent longer considering the card linked to the matching bias effect [14]. It is important to note, however, that longer response times may reflect other cognitive processes than deliberation [32].

Performance on some problems was associated with performance on other problems. That is, if someone correctly answered a problem in one of these sets, they were also highly likely to

correctly answer the other problems in the set. These sets were: (1) Lily Pads, Widgets and Petrol Station; (2) Hospitals, Four Cards and Two Drivers; (3) Birth Order and Coin Tosses. This is different with how these problems have been typically clustered *a priori* in the research literature: (I) Lily Pads, Widgets and Bat and Ball (CRT); (II) Hospitals and Two Drivers (explained below); (III) Hospitals, Birth Order and Coin Tosses (representativeness heuristic); (IV) Birth Order and Coin Tosses (probability theory). Consideration of these problem groupings follows.

Correctly answering all three problems in (I) entailed not being distracted by particular pieces of information in the problems so as to stay focused on uncovering the real underlying relationships. The Lily Pads and Widget problems can mislead if attention is over focused on the numbers, and conversely, the Petrol Station problem can mislead if there is too much focus on the idea of a discount. While the Lily Pads and Widget problems are traditionally paired with the Bat and Ball problem in the CRT, it may be that performance on the Bat and Ball problem did not appear as part of this set due to an added level of difficulty. With the problems in (I), avoiding being distracted by certain parts of the questions at the expense of others almost leads directly to the correct answer. However, with the Bat and Ball problem, further steps in mathematical reasoning still need to occur in answering which two numbers add together to give a result while also subtracting one from the other for another.

With the problems in (II) it is of interest that the Two Drivers problem was created specifically to be paired with the Hospitals problem to test for motivation in problem solving [23]. Within this framework further transparent versions of these problems were successfully devised to manipulate for difficulty. The Two Drivers problem was amended to have Driver B travelling at exactly 5 mph during the first half of the race and at exactly 95 mph during the last half of the race. The Hospitals problem was amended so the smaller hospital would have "only 2" babies born each day and where for a period of one year the hospitals recorded the number of days on which *all* of the babies born were boys. Could the association in (II) be pointing to how participants overcome initial fictitious mathematical rules? Maybe they reframe the question in simpler terms to see the pattern. The Four Cards Problem also elicited a high number of incorrect answers where, associated with mathematical training, the standard incorrect solution was avoided for more cognitively elaborate ones. Indeed, a gradation effect appeared across the groups where the standard error of the "D and 3" cards becomes "D only" (Table 4). Adrian Simpson and Derrick Watson found a comparable result across their two groups [14 p61]. This could again be pointing to having avoided an initial fictitious rule of simply concentrating on items directly found in the question, participants then seek to reframe the question to unearth the logical rule to be deduced. An added level of difficulty with this question may be why participants become trapped in a false answer. The eye-movement tracking study mentioned above supports this theory.

The problems in (III) fit naturally together as part of basic probability theory, a topic participants would have assimilated, or not, as part of various education curricula. While the equal likelihood of all possible outcomes with respect to a coin toss may be culturally assimilated, the same may not be as straightforward for birth gender outcomes where such assumptions could

**Table 4. Error distribution on the Four Cards Problem as a function of group.**

|  | Introductory | Standard | Advanced1 | Advanced2 | Academic |
|---|---|---|---|---|---|
| **Errors Total N (%)** | **59 (95%)** | **26 (96%)** | **30 (85%)** | **27 (61%)** | **19 (63%)** |
| Standard Error | 49% | 42% | 23% | 30% | 11% |
| "D only" Error | 20% | 27% | 40% | 26% | 53% |
| Other Error | 31% | 31% | 37% | 44% | 37% |

be swayed by biological hypothesis or folk wisdom [33]. The gradation of the results in terms of mathematical training does not support this possibility.

The effect of training on performance accuracy was more obvious in some problems compared to others, and to some extent, this was related to the type of problem. For instance, most of the problems in which performance was related to training (Four Cards, CRT [Lily Pads, Widgets, Bat and Ball], Two Drivers, Jack looking at Anne) could be classed as relying on logical and/or critical thinking. The one exception was the Birth Order problems, which are probability related.

In contrast, two of the three problems in which training did not appear to have much impact on performance (Hospitals and Coin Tosses) require domain-specific knowledge. The Hospitals problem requires a degree of knowledge about sampling statistics. This is a topic of quite distinct flavour that not all mathematically trained individuals gain familiarity with. On the other hand, all groups having performed well on the Coin Tosses problem is in line with a level of familiarity with basic probability having been originally presented at high school. While the questioning of patterning as negatively correlated with randomness is similar to that appearing in the Birth Order question, in the Birth Order question this aspect is arguably more concealed. These results and problem grouping (III) could be pointing to an area for improvement in teaching where the small gap in knowledge required to go from answering the Coin Tosses problem correctly to achieving similarly with the Birth Order problem could be easily addressed. A more formal introduction to sampling statistics in mathematical training could potentially bridge this gap as well as further be extended towards improvement on the Hospitals problem.

The other problem where performance was unrelated to training, the Petrol Station problem, cannot be characterised similarly. It is more of a logical/critical thinking type problem, where there remains some suggestion that training may have impacted performance, as the Academic group seemed to perform better than the rest of the sample. An alternate interpretation of this result is therefore that this problem should not be isolated but grouped with the other problems where performance is affected by training.

Although several aspects of the data suggest mathematics training improves the chances that someone will solve problems of the sort examined here, differences in the performance of participants in the Advanced1, Advanced2 and Academic groups were not obvious. This is despite the fact that large differences exist in the amount of training in these three groups. The first two groups were undergraduate students and the Academic group all had PhDs and many were experienced academic staff. One interpretation of this result is current mathematics training can only take someone so far in terms of improving their abilities with these problems. There is a point of demarcation to consider in terms of mathematical knowledge between the Advanced1, Advanced2 and Academic groups as compared to the Introductory and Standard groups. In Australia students are able to drop mathematical study at ages 15–16 years, or choose between a number of increasingly involved levels of mathematics. For the university in this study, students are filtered upon entry into mathematics courses according to their current knowledge status. All our groups involved students who had opted for post-compulsory mathematics at high school. And since our testing occurred in second semester, some of the mathematical knowledge shortfalls that were there upon arrival were bridged in first semester. Students must pass a first semester course to be allowed entry into the second semester course. A breakdown of the mathematics background of each group is as follows:

1. The Introductory group's mathematics high school syllabus studied prior to first semester course entry covered: Functions, Trigonometric Functions, Calculus (Introduction to Differentiation, Applications of the Derivative, Antiderivatives, Areas and the Definite

Integral), Financial Mathematics, Statistical Analysis. The Introductory group then explored concepts in mathematical modelling with emphasis on the importance of calculus in their first semester of mathematical studies.

2. The Standard group's mathematics high school syllabus studied prior to first semester course entry covered: Functions, Trigonometric Functions, Calculus (Rates of Change, Integration including the method of substitution, trigonometric identities and inverse trigonometric functions, Areas and Volumes of solids of revolution, some differential equations), Combinatorics, Proof (with particular focus on Proof by Mathematical Induction), Vectors (with application to projectile motion), Statistical Analysis. In first semester their mathematical studies then covered a number of topics the Advanced1 group studied prior to gaining entrance at university; further details on this are given below.

3. The Advanced1 group's mathematics high school syllabus studied prior to first semester course entry covered: the same course content the Standard group covered at high school plus extra topics on Proof (develop rigorous mathematical arguments and proofs, specifically in the context of number and algebra and further develop Proof by Mathematical Induction), Vectors (3 dimensional vectors, vector equations of lines), Complex Numbers, Calculus (Further Integration techniques with partial fractions and integration by parts), Mechanics (Application of Calculus to Mechanics with simple harmonic motion, modelling motion without and with resistance, projectiles and resisted motion). The Standard group cover these topics in their first semester university studies in mathematics with the exclusion of further concepts of Proof or Mechanics. In first semester the Advanced1 group have built on their knowledge with an emphasis on both theoretical and foundational aspects, as well as developing the skill of applying mathematical theory to solve practical problems. Theoretical topics include a host of theorems relevant to the study of Calculus.

In summary, at the point of our study, the Advanced1 group had more knowledge and practice on rigorous mathematical arguments and proofs in the context of number and algebra, and more in-depth experience with Proofs by Induction, but the bulk of extra knowledge rests with a much deeper knowledge of Calculus. They have had longer experience with a variety of integration techniques, and have worked with a variety of applications of calculus to solve practical problems, including a large section on mechanics at high school. In first semester at university there has been a greater focus on theoretical topics including a host of theorems and associated proofs relevant to the topics studied. As compared to the Introductory and Standard groups, the Advanced1 group have only widened the mathematics knowledge gap since their choice of post-compulsory mathematics at high school. The Advanced2 group come directly from an Advanced1 cohort. And the Academics group would have reached the Advanced1 group's proficiency as part of their employment. So, are specific reasoning skills resulting from this level of abstract reasoning? Our findings suggest this should certainly be an area of investigation and links in interestingly with other research work. In studying one of the thinking tasks in particular (the Four Cards Problem) and its context of conditional inference more specifically, Inglis and Simpson [15] found a clear difference between undergraduates in mathematics and undergraduates in other university disciplines, yet also showed a lack of development over first-year university studies on conditional inference measures. A follow up study by Attridge and Inglis [22] then zeroed in on post-compulsory high school mathematical training and found that students with such training did develop their conditional reasoning to a greater extent than their control group over the course of a year, despite them having received no explicit tuition in conditional logic. The development though, whilst demonstrated as not being the result of a domain-general change in cognitive capacity or thinking disposition, and most likely associated

with the domain-specific study of mathematics, revealed a complex pattern of endorsing more of some inferences and less of others. The study here focused on a much broader problem set associated with logical and critical thinking and it too is suggestive of a more complex picture in how mathematics training may be contributing to problem solving styles. A more intricate pattern to do with the impact of mathematical training on problem solving techniques is appearing as required for consideration.

There is also a final interpretation to consider: that people in the Advanced 1, Advanced2 and Academic groups did not gain anything from their mathematics training in terms of their ability to solve these problems. Instead, with studies denying any correlation of many of these problems with what is currently measured as intelligence [30], they might still be people of a particular intelligence or thinking disposition to start with, who have been able to use that intelligence to not only solve these problems, but also survive the challenges of their mathematics training.

That the CRT has been traditionally used as a measure of baseline thinking disposition and that performance has been found to be immutable across groups tested is of particular interest since our results show a clear possible training effect on these questions. CRT is tied with a willingness to engage in effortful thinking which presents as a suitable ability for training. It is beyond the scope of this study, but a thorough review of CRT testing is suggestive of a broader appreciation and better framework to understand thinking disposition, ability and potential ability.

Mathematical training appears associated with certain thinking skills, but there are clearly some subtleties that need to be extricated. The thinking tasks here add to the foundational results where the aim is for a firmer platform on which to eventually base more targeted and illustrative inquiry. If thinking skills can be fostered, could first year university mathematics teaching be improved so that all samples from that group reach the Advanced1 group level of reasoning? Do university mathematics courses become purely about domain-specific knowledge from this point on? Intensive training has been shown to impact the brain and cognition across a number of domains from music [34], to video gaming [35], to Braille reading [36]. The hypothesis that mathematics, with its highly specific practice, fits within this list remains legitimate, but simply unchartered. With our current level of understanding it is worth appreciating the careful wording of the NYU Courant Institute on 'Why Study Math?' where there is no assumption of causation: "Mathematicians need to have good reasoning ability in order to identify, analyze, and apply basic logical principles to technical problems." [37].

## Limitations

One possible limitation of the current study is that the problems may have been too easy for the more advanced people, and so we observed a ceiling effect (i.e., some people obtained 100% correct on all problems). This was most obvious in the Advanced1, Advanced2 and Academic groups. It is possible that participants in these groups had developed logical and critical thinking skills throughout their mathematical training that were sufficient to cope with most of the problems used in this study, and so this would support the contention that training in mathematics leads to the development of logical and critical thinking skills useful in a range of domains. Another interpretation is that participants in these groups already possessed the necessary thinking skills for solving the problems in this study, which is why they are able to cope with the material in the advanced units they were enrolled in, or complete a PhD in mathematics and hold down an academic position in a mathematics department. This would then suggest that training in mathematics had no effect on abstract thinking skills—people in this study possessed them to varying extents prior to their studies. This issue might be settled in a future

study that used a greater number of problems of varying difficulties to maximise the chances of finding a difference between the three groups with the most amount of training. Alternatively, a longitudinal study that followed people through their mathematics training could determine whether their logical and critical thinking abilities changed throughout their course.

A further limitation of the study may be that several of the reasoning biases examined in this study were measured by only one problem each (i.e., Four Cards Problem, Two Drivers, Petrol Station, Jack looking at Anne). A more reliable measure of these biases could be achieved by including more problems that tap into these biases. This would, however, increase the time required of participants during data collection, and in the context of this study, would mean a different mode of testing would likely be required.

## Conclusion

Broad sweeping intuitive claims of the transferable skills endowed by a study of mathematics require evidence. Our study uniquely covers a wide range of participants, from limited mathematics training through to research academics in the mathematical sciences. It furthermore considered performance on 11 well-studied thinking tasks that typically elude participants in psychological studies and on which results have been uncorrelated with general intelligence, education levels and other demographic information [15, 16, 30]. We identified different performances on these tasks with respect to different groups, based on level of mathematical training. This included the CRT which has developed into a method of measuring baseline thinking disposition. We identified different distributions of types of errors for the mathematically trained. We furthermore identified a performance threshold that exists in first year university for those with high level mathematics training. This study then provides insight into possible changes and adjustments to mathematics courses in order for them to fulfil their advertised goal of reaching improved rational and logical reasoning for a higher number of students.

It is central to any education program to have a clear grasp of the nature of what it delivers and how, but arguably especially so for the core discipline that is mathematics. In 2014 the Office of The Chief Scientist of Australia released a report "Australia's STEM workforce: a survey of employers" where transferable skills attributed to mathematics were also ones that employers deemed as part of the most valuable [38]. A better understanding of what mathematics delivers in this space is an opportunity to truly capitalise on this historical culture-crossing subject.

## Supporting information

**S1 Data.**
(XLSX)

## Acknowledgments

The authors would like to thank Jacqui Ramagge for her proof reading and input, as well as support towards data collection.

## Author Contributions

**Conceptualization:** Clio Cresswell, Craig P. Speelman.

**Data curation:** Clio Cresswell, Craig P. Speelman.

**Formal analysis:** Clio Cresswell, Craig P. Speelman.

**Investigation:** Clio Cresswell, Craig P. Speelman.

**Methodology:** Clio Cresswell, Craig P. Speelman.

**Project administration:** Clio Cresswell.

**Resources:** Clio Cresswell.

**Writing – original draft:** Clio Cresswell, Craig P. Speelman.

**Writing – review & editing:** Clio Cresswell, Craig P. Speelman.

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
