## [Decision Letter · Decision Letter 0]

17 Mar 2020

PONE-D-20-01159

Does mathematics training lead to better logical thinking and reasoning? A cross-sectional assessment from students to professors

PLOS ONE

Dear Professor Speelman,

Thank you for submitting your manuscript to PLOS ONE. I have sent it to two expert reviewers and have received their comments back. As you can see at the bottom of this email, both reviewers are positive about your manuscript but raise some issues that you would need to address before the manuscript can be considered for publication. Notably, reviewer #1 points out that the manuscript should include a discussion on the reasons why individuals with math training may have improved reasoning skills (e.g., logical intuitions versus deliberate thinking). The reviewer also rightly mentions that your sample sizes are limited, notably for the most advanced groups. This should be discussed and acknowledged. Reviewer #2 has a number of conceptual and methodological points that you will also have to address. The reviewer provides very thorough comments and I will not reiterate the points here. However, note that both reviewers suggest that you need to improve the figures and I agree with them.   

We would appreciate receiving your revised manuscript by May 01 2020 11:59PM. To enhance the reproducibility of your results, we recommend that if applicable you deposit your laboratory protocols in protocols.io, where a protocol can be assigned its own identifier (DOI) such that it can be cited independently in the future. For instructions see: http://journals.plos.org/plosone/s/submission-guidelines#loc-laboratory-protocols

Please include the following items when submitting your revised manuscript:
A rebuttal letter that responds to each point raised by the academic editor and reviewer(s). This letter should be uploaded as separate file and labeled 'Response to Reviewers'.A marked-up copy of your manuscript that highlights changes made to the original version. This file should be uploaded as separate file and labeled 'Revised Manuscript with Track Changes'.An unmarked version of your revised paper without tracked changes. This file should be uploaded as separate file and labeled 'Manuscript'.

We look forward to receiving your revised manuscript.

Kind regards,

Jérôme Prado

Academic Editor

PLOS ONE

Journal Requirements:

2. Please include additional information regarding the survey or questionnaire used in the study and ensure that you have provided sufficient details that others could replicate the analyses. For instance, if you developed a questionnaire as part of this study and it is not under a copyright more restrictive than CC-BY, please include a copy, in both the original language and English, as Supporting Information. Please also let us know if it would be possible to provide the anonymized data points necessary to replicate the statistical analyses, for instance, as shown in fig 1 and 2. If so, please deposit those to a suitable data repository or include them in the Supporting Information files.

Please provide an amended Funding Statement that declares *all* the funding or sources of support received during this specific study (whether external or internal to your organization) as detailed online in our guide for authors at http://journals.plos.org/plosone/s/submit-now.  Please state what role the funders took in the study.  If any authors received a salary from any of your funders, please state which authors and which funder. If the funders had no role, please state: "The funders had no role in study design, data collection and analysis, decision to publish, or preparation of the manuscript."

Reviewers' comments:

Reviewer #1: I think this is a very good and interesting manuscript trying to answer an important research question. I propose some changes that I believe should be applied before publication.

1. Each reasoning bias is measured with only one problem. In reasoning research, it is rather common to measure each type of reasoning problem with a series of structurally equivalent reasoning problems, so the results will be independent of contexts effects and will be generalizable to that type of problem. Here, the authors only measured each reasoning bias with one single problem and this might be problematic (see, for example: Fiedler & Hertel, 1994). I think this can be addressed by simply discussing it in the limitation section.

2. This is rather a minor issue, but the discussion on the CRT problems is not up-to-date (page 7). Most recent experiments on dual process theory suggest that people who are able to correctly solve these reasoning problems (including the CRT) do so intuitively, and not because they engaged in careful deliberation (Bago & De Neys, 2019). Intelligence made people have better intuitive responses (Thompson, Pennycook, Trippas & Evans, 2018). Similarly, this problems persists in the discussion about reaction times (page 25). Longer reaction times does not necessarily mean that people engaged in deliberation (see: Evans, Kyle, Dillon & Rand, 2015). Response time might be driven by decision conflict or response rationalization. These issues could be clarified with some changes in the wording or some footnotes on page 7 and 25. Furthermore, it would be interesting to have a discussion on how mathematical education helps people overcome their biases. Is it because it creates better intuition, or helps people engage in deliberation? An interesting question this manuscript does not discuss. It’s on the authors whether or not they discuss this latter point now, but the changes on page 7 and 25 should be made.

3. A more serious problem is the rather small sample size (especially in the more advanced groups). This small sample size makes the appearance of both false negatives and false positives more likely. Perhaps, the authors could compute the Bayes Factors for the chi-square or logistic regression test, so we can actually see how strong the evidence is for or against the null. This is especially important as the authors run a great number of explorative analysis (Table 3), and some of those results might need to be interpreted with great caution (depending on the Bayes Factor).

Other:

The graphs are not looking good, they should comply with APA formatting. At the very least, the axis titles should be meaningful and measure units should be written there.

The presentation order of the problems is quite unusual; why isn’t it random? Why did the authors decide on this order?

Reviewer #2: The study reported in this paper compared five groups of participants with varying levels of mathematical expertise on a set of reasoning tasks. The study is interesting and informative. It extends the current literature on this topic (which is reviewed very nicely in the introduction). However, there are some issues with the current analysis and interpretation that should be resolved prior to publication. I have therefore recommended major revisions. My comments are organised in the order in which they came up in the paper and they explain my responses to the questions above.

1. Line 114 – “general population” a bit misleading – they were also students but from other disciplines.

2. Line 124 onwards reads:

“The ultimate question to consider here is: are any skills associated with mathematics training innate or do they arise from skills transfer? Though to investigate how mathematical training affects reasoning skills, randomised sampling and randomised intervention to reveal causal relationships are clearly not viable. With so many possible confounding variables and logistical issues, it is even questionable what conclusions such studies might provide. Furthermore, a firm baseline from which to propose more substantive investigations is still missing.”

I find this paragraph slightly problematic because the current study doesn’t inform us on this ultimate question, so it makes the outline of the current study in the following paragraph feel unsatisfactory. I think the current study is important but prefacing it with this paragraph underplays that importance. And I think a randomised controlled study, although not viable, would give the answers we need because the random allocation to groups would allow us to rule out any confounding variables. Finally, the last sentence in this paragraph is unclear to me.

3. In the descriptions of the five participants groups the authors refer to the group’s level of interest in mathematics, but this seems like an overgeneralisation to me. Surely the introductory group could contain a biology student who also happens to be good at mathematics and very much enjoy it? I would be more comfortable with the descriptions if the parts about interest level were removed.

4. How many of the 123 first year students were in each of the three first year groups?

5. Line 313 – the standard group is referred to as “university mathematics students”, but they are not taking mathematics degreed.

6. Line 331 - what is a practice class?

7. Were the data collection settings quiet? From the description it sounds like groups of participants were completing the study at the same time in the same room, but the authors should make this explicit for the sake of the method being reproducible. E.g. how many students were in the room at the time?

8. Line 355-356 – the authors should not use the term “marginally worse” because this is statistically inappropriate – in a frequentist approach results are either significant or non-significant.

9. Line 340 – “approximate completion times were noted.”

This doesn’t sound rigorous enough to justify analysing them. Their analysis is interesting, but the authors should remind readers clearly whenever the response times are analysed or discussed that their recording was only manual and approximate.

10. I suggest replacing Figure 1 with a bar chart showing standard error of the mean on the error bars. A table with mean score out of 11 and the standard deviation for each group may also be useful. Figure 2 should be a scatterplot rather than a box and whisker plot.

11. Was the 0-11 total correct score approximately normally distributed across the full sample?

12. Chi square analysis requires at least 5 cases in each cell, was this met? It seems not since Table 1 shows lots of cells in the “no response” row having 0% of cases.

13. The chi-square analyses should be followed up with post hoc tests to see exactly where the differences between groups are. The descriptions as they stand aren’t that informative (as readers can just look at Table 1) without being backed up by post hoc tests.

14. For each chi square analysis in the text, I would find it easier to read if the test statistics came at the top of the paragraph, before the description.

15. Line 381-383 – “Of note, also, is the relatively low proportion of those in the higher training groups who, when they made an error, did not make the standard error, a similar result to the one reported by Inglis and Simpson [11]."

I think this is supposed to say that a low proportion did make the standard error or that a high proportion did not make the standard error.

16. Line 403 - p values this small should be reported as p < .001 rather than p = .000 since they aren’t actually 0.

17. Line 476 – “…if a particular outcome variable was predicted significantly by a particular predictor variable, the converse relationship was also observed”

Isn’t that necessarily the case with regression analyses, like with correlations?

18. I don’t think the logistic regression analyses add much to the paper and at the moment they come across as potential p-hacking since they don’t clearly relate to the research question. To me they make the paper feel less focused. Having said that, there is some interesting discussion of them in the Discussion section. I’d recommend adding some justification to the introduction for why it is interesting to look at the relationships among tasks (without pretending to have made any specific hypotheses about the relationships, of course).

19. Line 509 would be clearer if it read “between these groups and the introductory and standard groups”

20. Lines 597 – 620 - This is an interesting discussion, especially the suggestion that advanced calculus may be responsible for the development. No development in reasoning skills from the beginning of a mathematics degree onwards was also found by Inglis and Simpson (2009), who suggested that the initial difference between mathematics and non-mathematics undergraduates could have been due to pre-university study of mathematics. Attridge & Inglis (2013) found evidence that this was the case (they found no difference between mathematics and non-mathematics students at age 16 but a significant difference at the end of the academic year, where the mathematics students had improved and the non-mathematics students had not).

Could the authors add some discussion of whether something similar may have been the case with their Australian sample? E.g. do students in Australia choose whether, or to what extent, to study mathematics towards the end of high school? If not, the description of the groups suggests that there were at least differences in high school mathematics attainment between groups 1-3, even if they studied the same mathematics curriculum. Do the authors think that this difference in attainment could have led to the differences between groups in the current study?

21. Line 617 – “Intensive training has been shown to impact the brain and cognition across a number of domains from music, to video gaming, to Braille reading [31].”

Reference 31 appears to only relate to music. Please add references for video gaming and Braille reading.

22. I recommend editing the figures from SPSS’s default style or re-making them in Excel or DataGraph to look more attractive.

23. I cannot find the associated datafile anywhere in the submission. Apologies if this is my mistake.

---

## [Author Response · Author response to Decision Letter 0]

20 Apr 2020

All responses are detailed against the specific reviewers' comments in the Response to Reviewers document

---

## [Decision Letter · Decision Letter 1]

11 Jun 2020

PONE-D-20-01159R1

Does mathematics training lead to better logical thinking and reasoning? A cross-sectional assessment from students to professors.

PLOS ONE

Dear Dr. Speelman,

Thank you for submitting your revised manuscript to PLOS ONE. I have sent it to reviewer #2 and have now received the reviewer's comment. As you can see, the reviewer thinks that the manuscript is improved but has some outstanding issues that you would need to address in another round of revision. I notably agree with the reviewer that you should provide the raw data, allowing readers to replicate your analyses. Therefore, I invite you submit a revised version of your manuscript.

We look forward to receiving your revised manuscript.

Kind regards,

Jérôme Prado

Academic Editor

PLOS ONE

Reviewers' comments:

Reviewer #2: The manuscript has improved but there are still a few issues that should be resolved prior to publication.

1. On lines 96, 97, 100 and 102, the references to “general population” should be changed to reflect the fact that these participants were non-mathematics (arts) students.

2. Line 306 – change “mathematics students” to “university students”.

3. The method section doesn’t specify the gender split and mean age of the sample.

4. Table 3 - values the p values listed as .000 should be changed to <.001.

5. Table 3 - I suggest repeating the list of problem numbers and names in the legend. It may make for a long legend but would make it much easier for the reader to interpret the table.

6. I am not sure what the new post hoc tests are comparing. What I expected was to see group 1 compared to groups 2, 3, 4 and 5, and so on. This would tell us which groups are statistically different from each other. At the moment we only know from the overall chi square tests whether there are any differences among the groups or not, we don’t know specifically which groups are statistically different from each other and which ones are not. We only have the authors’ interpretations based on the observed counts.

7. Line 584 - change “performance was correlated with training” to “performance was related to training” to avoid any confusion since a correlation analysis was not performed.

8. Data file – I had expected the data file to give the raw data rather than summary data, i.e. with each participant in a separate row, and a column indicating their group membership, a column giving their age, a column for sex etc (including all the demographics mentioned in the method), and a column for each reasoning question. This would allow other researchers to replicate the regression analyses and look at other relationships within the dataset. Without being able to replicate all analyses in the paper, the data file does not meet the minimal data set definition for publication in PLOS journals: https://journals.plos.org/plosone/s/data-availability.

---

## [Author Response · Author response to Decision Letter 1]

16 Jun 2020

Please see "Response to Reviewers" document

---

## [Editor Report · Decision Letter 2]

1 Jul 2020

Does mathematics training lead to better logical thinking and reasoning? A cross-sectional assessment from students to professors.

PONE-D-20-01159R2

Dear Dr. Speelman,

We’re pleased to inform you that your manuscript has been judged scientifically suitable for publication and will be formally accepted for publication once it meets all outstanding technical requirements.

Kind regards,

Jérôme Prado

Academic Editor

PLOS ONE
---

## [Editor Report · Acceptance letter]

7 Jul 2020

PONE-D-20-01159R2 

Does mathematics training lead to better logical thinking and reasoning? A cross-sectional assessment from students to professors. 

Dear Dr. Speelman:

I'm pleased to inform you that your manuscript has been deemed suitable for publication in PLOS ONE. Congratulations! Your manuscript is now with our production department. 

Kind regards, 

on behalf of

Dr. Jérôme Prado 

Academic Editor

PLOS ONE